# Modeling and Performance Analysis of LBT-Based RF-Powered NR-U Network for IoT

**DOI:** 10.3390/s24165369

**Published:** 2024-08-20

**Authors:** Varada Potnis Kulkarni, Radhika D. Joshi

**Affiliations:** Department of Electronics and Telecommunication Engineering, COEP Technological University, Formerly College of Engineering Pune, Wellesley Road, Shivajinagar, Pune 411005, India; rdj.extc@coeptech.ac.in

**Keywords:** LBT, Markov chain, NR-U, green, cellular IoT, RF-EH, RF-powered

## Abstract

Energy harvesting combined with spectrum sharing offers a promising solution to the growing demand for spectrum while keeping energy costs low. New Radio Unlicensed (NR-U) technology enables telecom operators to utilize unlicensed spectrum in addition to the licensed spectrum already in use. Along with this, the energy demands for the Internet of Things (IoT) can be met through energy harvesting. In this regard, the ubiquity and ease of implementation make the RF-powered NR-U network a sustainable solution for cellular IoT. Using a Markov chain, we model the NR-U network with nodes powered by the base station (BS). We derive closed-form expressions for the normalized saturated throughput of nodes and the BS, along with the mean packet delay at the node. Additionally, we compute the transmit outage probability of the node. These quality of service (QoS) parameters are analyzed for different values of congestion window size, TXOP parameter, maximum energy level, and energy threshold of the node. Additionally, the effect of network density on collision, transmission, and energy harvesting probabilities is observed. We validate our model through simulations.

## 1. Introduction

Communication among humans, machines, and between humans and machines is growing exponentially. The ever-increasing number of things connected to the internet and the corresponding rise in exciting and useful applications pose further challenges to the design and development of communication networks. The Internet of Things (IoT) is evolving into the Internet of Everything (IoE). The number of IoT devices will reach more than USD 39 billion by 2033, marking more than a twofold increase over the next nine years, as reported by https://www.statista.com (accessed on 8 August 2024) [1].

Also, the number of IoT devices per person was 3.6 on a worldwide average in the year 2023 according to a survey [2]. To cater to the increased demand for data rates and the number of devices connected to the network, fifth-generation (5G) and beyond fifth-generation (B5G) communication technologies are being standardized. Novel techniques are being developed to simultaneously support high bandwidth, low latency, high reliability, and fairness among incumbent technologies.

To meet the growing demand for data rates and to support the massive number of connections, it is essential to utilize all the spectrum types as described by Qualcomm Technologies [3]. There are mainly three spectrum types: (a) licensed spectrum (exclusive use, e.g., over 40 bands of long-term evolution used globally), (b) shared spectrum with new paradigms (e.g., 2.3 GHz in Europe and 3.7 GHz in the USA), and (c) unlicensed spectrum (shared use, 2.4 GHz/5–7 GHz/57–71 GHz). This paper focuses on New Radio Unlicensed (NR-U) technology, which involves the use of an unlicensed spectrum by cellular operators. It represents a fifth-generation (5G) extension to License-Assisted Access (LAA) used in 4G communications and standardized by the Third-Generation Partnership Project (3GPP) [4]. NR-U coexists with other technologies using unlicensed spectra such as Wi-Fi and Bluetooth. It includes a Listen Before Talk (LBT) mechanism for multiple access, which is similar to the Carrier Sense Multiple Access (CSMA) used in Wi-Fi [5]. In the LBT protocol, nodes willing to transmit data ‘sense’ the carrier before sending the data and adjust their backoff depending on the congestion level. NR-U is widely used in various sectors involving IoT networks, as described by Y. Liu et al. [6]. The use of NR-U in the oil and gas industry is discussed by A.H.S et al. [7].

IoT networks with high data rates and large numbers of users require significant energy for their operations. The energy harvesting feature helps maintain the sustainability of the network, as described by N Ansari et al. [8]. In energy harvesting communication networks, various types of harvesting technologies can be incorporated. The base stations can be equipped with renewable power supplies such as solar power or wind power. Smartphones and other portable devices can harvest energy from thermal changes in the environment, piezoelectric effects, and RF energy present in the ambiance, as described by the authors [9]. RF energy is widely available and can be harvested cost-effectively, hence, RF-powered IoT networks are popular, as mentioned by M. A. Abd-Elmagid et al. [10]. These networks are used in applications such as smart farming [11], museum ambiance control [12], and segments such as civil infrastructure and manufacturing [13].

There are various ways in which a wireless network can be RF-powered. Inductive and magnetic coupling are useful for very short distances, ranging up to a few meters. For longer distances up to a few kilometers, the far-field region of the propagation wave can be used, as described by Niyato et al. [14]. In this technique, RF power from cellular base stations (BS), wireless access points, as well as radio and television transmitters is used. Due to the ubiquity of cellular systems, this work considers a base station-powered RF network. There are different ways in which a base station can power IoT nodes, e.g., using the duty cycle method for data and energy transfer, polling nodes for energy harvesting, or employing CSMA/LBT-based mechanisms. In order to combine the benefits of spectrum sharing and cellular systems, we consider an RF-powered IoT network that uses NR-U protocol for communication in which LBT is mandated for transmission. Thus, LBT-based RF power transfer is a natural choice.

In this paper, the base station is powered by traditional non-energy harvesting methods, and other nodes in the network harvest RF energy from BS during the data transfer. In particular, we mathematically model the LBT-based RF-powered network and analyze the performance of the nodes in the network. Using the expressions provided by the model, we analyze the node throughput, delay, and outage probability. The node throughput is specifically useful for enhanced multimedia broadband (eMBB) applications of 5G and 6G networks, and the mean delay of a node is significant in the context of ultra-reliable low latency communications (URLLC).

This paper is organized as follows: in Section 2, we present the related work regarding energy harvesting communication networks as well as a literature survey of mathematical modeling attempts. We describe the system model in Section 3. In Section 4, we present the performance analysis of the quality of service (QoS) parameters. We check the validity of the mathematical model with simulations in Section 5 and present the conclusions in Section 6. In Section 7, we discuss future work.

## 2. Literature Survey

The algorithms and protocols designed for wireless networks work differently when nodes harvest energy and work under energy constraints. Also, at the same time, the design should meet the criteria of 5G/6G networks. In [15], Huang et al. proposed an architecture for IoT applications in 5G communications that incorporates software-defined network management and energy harvesting nodes with a mobile charger. The paper describes the management of both uplink and downlink energy transfers in multi-hop wireless sensor networks, assisted by a mobile charger capable of replenishing energy for nodes. The authors also note that next-generation communication systems will likely be a combination of advanced technologies, highlighting the need to integrate these technologies and optimize them for better efficiency. In another study, the eHealth application involving wearable devices is modeled using the Poisson cluster process and analyzed for the probability that a wearable device correctly notifies the message along with other QoS parameters [16] by P-V Mekikis et al. The wearable devices are wirelessly charged.

There are relatively few papers on the modeling and analysis of energy harvesting solutions for 5G and B5G systems. In [17], Di Zhai et al. proposed two multi-user scheduling schemes for IoT networks with wireless-powered nodes and a hybrid access point capable of transferring wireless power. The first policy attempts to achieve maximum system throughput in the absence of channel state information, considering residual energy at nodes. The authors consider the accumulate and transmit protocol. The second policy [17] attempts to obtain fairness for nodes with poor channel quality. The delay constraints as well as spectrum sharing are not considered. The authors in [18] presented a Markovian model for wireless sensor networks where nodes can be chosen between unlicensed and licensed bands. They also provided a Markov decision process formulation to derive an optimal transmission scheduling policy. In this model, energy harvesting is not considered, as the nodes are powered by traditional methods.

We briefly describe the analytical models for CSMA, which are used by LBT. The homogenous coexistence consisting of only NR nodes in the absence of Wi-Fi in the NR-U network was considered [19] by S. Muhammad. The throughput, delay, and collision probability were calculated for different priorities of nodes in a multiclass environment. The model was applied to the 5G NR-U-enabled ICU hospital setup. The nodes were non-energy harvesting (non-EH). The modeling and analysis of WLAN with RF-powered nodes were performed [20] by Y. Zhao et al. The access point (AP) was grid-powered while the stations harvested RF energy from the AP when the AP transmitted data to other nodes. The network was modeled using a three-dimensional Markov chain. The uplink and downlink throughput were calculated for AP as well as a station. In another attempt, for general energy harvesting nodes without dependence on AP, modeling and analysis were performed [21]. The stations harvested energy, modeled as the Poisson process. As the number of slots required for sufficient charging of stations was assumed to be large compared to the average backoff duration, the throughput and delay calculated were independent of other parameters such as congestion window size and the number of stations. The Markov chain analysis performed in [19,20,21] is based on the classic Bianchi model [22]. The literature presents models for RF-powered WLAN to compute throughput and delay. In addition to these parameters, the outage probability is an important metric for energy-harvesting wireless networks. Most of the work in the literature is dedicated to finding outage probability, considering the signal-to-interference noise ratio (SINR) at the receiver’s end, e.g., by R Vaze [23]. Reference [24] computed and analyzed the transmit outage probability.

This paper considers an IoT network that is RF-powered by BS. The network operates under the NR-U protocol using the LBT feature and is modeled using a three-dimensional Markov chain. Closed-form expressions for the collision probability of BS and nodes, as well as transmission probability, energy harvesting probability, and throughput, are calculated. The expression for the mean delay of a packet at a node is also derived. The outage probability of the node is computed. These QoS parameters are analyzed for the size of the network, the size of the congestion window, the TXOP value, the maximum energy level, and the energy threshold of the node. While there are numerous attempts in the literature to model RF-powered WLAN, to the best of our knowledge, this is the first attempt to model and analyze an RF-powered NR-U network for IoT communication. The model helps us to analyze the key QoS parameters for NR-U-specific parameters such as TXOP. In [19], the NR-U network with non-EH nodes is analyzed for collision probability, throughput, and mean delay. In this work, we analyze the RF-powered NR-U network for collision, transmission, and energy harvesting probabilities, as well as for throughput and delay. Also, we compute the transmit outage probability of RF-powered LBT, which adds novelty to the existing work. The authors’ contribution is to envision the NR-U network with RF energy harvesting and comprehensively analyze it for QoS parameters, which include throughput, delay, and transmit outage.

## 3. System Model

This section is organized as follows: Section 3.1 describes the model and the assumptions. In Section 3.2, the introduction to the LBT protocol is provided. Section 3.3 describes the Markov chain modeling. In Section 3.4, the resultant formulae of QoS parameters are provided.

### 3.1. Model and Assumptions

In this paper, we consider an NR-U network where nodes/user equipment (UE) harvest RF energy from the access point or BS. The IoT nodes function equivalently to UE. The nodes harvest energy when the BS sends data to other nodes. As standardized in 3GPP, the nodes and the BS follow the LBT protocol for random access [4]. For simplification, we only consider the NR-U nodes and do not consider the existence of Wi-Fi networks and other incumbent technologies such as Bluetooth. This can also be referred to as homogenous coexistence [19]. It applies to scenarios where other technologies do not exist in the area considered, or some form of multiplexing—either spatial or temporal—is used to avoid interference among these different technologies.

The node is equipped with an RF energy harvester. A typical node with an RF energy harvester is shown in Figure 1.

The RF energy harvester consists of an antenna, a matching circuit, a rectifier, and a load or energy storage device [25]. Various types of RF energy harvesters—according to bandwidths—are available, namely single-band, multi-band, and wide-band [26]. For the system model under consideration, single-band RF energy harvesters for industrial, scientific, and medical (ISM) bands are suitable. There is substantial literature available on rectenna design for ISM bands (2.4 GHz and 5.8 GHz) [26]. For example, Khan et al. designed a single-band rectenna that generates 2.75 V at a distance of 10 meters [27] by Khan et al.

We consider a base station with a transmit power of *Q* watts located at the center of a circular area with a radius of *R* meters. The system model is shown in Figure 2.

The nodes are randomly present in the circular area. The nodes harvest sufficient energy to execute the NR-U protocol. The base station is powered by the grid and is always ON while the nodes are considered RF-powered. The nodes harvest RF power from BS when the BS transmits data to other nodes. The energy is stored in the supercapacitor. It is assumed that in one successful frame transmission of the BS, nodes harvest one unit of energy. For the transmission of one frame, the entire stored energy is used.

### 3.2. Introduction to LBT in RF-Powered NR-U

The flowchart of the LBT procedure followed by an RF-powered node is provided in Figure 3.

When the nodes and BS have data to transmit, they wait for the clear channel assessment (CCA) duration. The nodes detect the energy level in the channel. According to 3GPP TR38.889 [4], the energy detection threshold is −73 dBm. If the channel is sensed to be free and the energy of the node is sufficient (i.e., greater than the threshold $C_{th}$), then the node draws a random value for backoff from the congestion window size, denoted by CW. The notations used in the paper are summarized in Table 1. The minimum value of the congestion window is denoted by $CW_{min}$ and the maximum value is denoted as $CW_{max}$. The node with the least backoff counter transmits first, and others try in the next frame duration. If a node’s energy is less than the threshold, then the nodes wait and will transmit when the energy becomes sufficient. The BS always has the energy to transmit as it is powered by the grid. If two or more nodes transmit in the same frame duration, a collision occurs. A collision can also happen when a node transmits simultaneously with the BS. The collision window is doubled each time a collision occurs.

The potential protocol stack of the envisioned system is shown in Figure 4.

The layers in the protocol stack of the NR-U base station and the IoT node using the NR-U framework are shown. Both the NR-U base station and the RF-powered node include an LBT manager in addition to other modules similar to New Radio 5G. A detailed description of the 5G New Radio architecture is available in 3GPP documentation [28]. Also, the variations in the protocol stack for NR-U are provided [29] by M. Hirzallah et al. The PHY and MAC layer modules perform functions for physical layer aspects and medium access control, respectively. They communicate with the scheduler in the NR-U base station, which in turn communicates with the LBT manager for channel sensing and decision-making based on the signal level. The scheduler is specific to the BS for coordinating the nodes in the network. The radio link control (RLC) layer ensures reliable data delivery. The packet data convergence protocol (PDCP) provides data compression for user and control data. The service data adaption protocol (SDAP) layer is responsible for the QoS framework of 5G. The functions of the radio resource control (RRC) layer include the establishment, maintenance of radio resources, and security management. In addition to these NR-U protocol layers, a power manager is essential in the architecture of an RF-powered IoT node, which performs the following functions: if the energy is available above the threshold, then signaling—like non-EH NR-U—takes place. If the energy is less than the threshold, then the node waits until its energy level is above the threshold by recharging itself with BS power.

### 3.3. Markov Chain Modeling

Markov chain modeling is a powerful mathematical tool used to compute steady-state probabilities, leading to the analysis of QoS parameters in communication networks. Here, we model the sequence of events in the LBT procedure followed by the RF-powered NR-U node using a discrete-time Markov chain (DTMC).

Let the number of stations be denoted as *N*. The initial congestion window size is $W_{0}$. The maximum number of times the window size can be doubled is denoted as *M*. The maximum charging level of the supercapacitor is denoted by $C_{max}$. Let m(t) denote the number of times the congestion window doubled (retransmission attempts), b(t) denote the backoff counter value, and c(t) denote the charging level of supercapacitor at time *t*. Then {m(t),b(t),c(t)} is a discrete-time stochastic process with the state space described in Figure 5. Let $P_{e}$ denote the probability that a node successfully harvests energy from the BS. Let $\tau_{N}$ be the probability that a node attempts a transmission, and $P_{c-N}$ denote the probability that a node collides with another node. Let $P_{c-BS}$ denote the probability that the BS encounters a collision and let the transmission probability of the $\tau_{BS}$. The five categories of state space are described as follows:Category 1: In this category, the backoff counter b(t) is decremented by one in every slot, but it does not reach zero, so the node cannot transmit. It harvests energy with the probability $P_{e}$ and the charging level is increased by one.Category 2: As the charging level of the node is full, the only change in the state space is that the backoff counter is decremented by one for this set of states.Category 3: In this category, the backoff counter is zero, and the node harvests energy with probability $P_{e}$ when BS transmits.Category 4: The node transmits as the backoff counter reaches zero. The collision occurs with the probability $P_{c-N}$ and the next value of the backoff counter is chosen with uniform probability from the congestion window size $W_{m+1}$. If the transmission is successful, then the congestion window is reset to size $W_{0}$.Category 5: The transitions are similar to that of Category 4. The difference is that the retransmission attempts reach the maximum value of *M*. From the state space transitions, it can be seen that {m(t),b(t),c(t)} is a DTMC. Here, it is important to consider three variables simultaneously so that the sequence of events under consideration follows the Markov property [30]. Therefore, the Markov chain is three-dimensional.

The Markov chain under consideration is too complex to obtain closed-form expressions by solving π=πP where *P* is the transition probability matrix. The steady-state distribution can be obtained either by solving numerically or using the decoupling technique [20]. The closed-form expressions after using the decoupling approximation are enlisted in Section 3.4.

### 3.4. QoS Parameters

The transmission probability of a node $\left(\tau_{N}\right)$, the collision probability of a node $\left(P_{c-N}\right)$, and the energy harvesting probability of a node $\left(P_{e}\right)$ can be found using the Markov chain model from Section 3.3. These probabilities depend on the collision probability and transmission probability of the BS. The expressions for $P_{c-BS}$ and $\tau_{BS}$ are obtained from the analysis involving non-EH nodes initially derived [22] and used later in other works, including [20]. The collision of a BS occurs when at least one node transmits along with the BS. Hence, $P_{c-BS}$ is expressed in (Equation 1) as follows: (1)$$P_{c-BS}=1-{\left(1-\tau_{N}\right)}^{N}.$$

The transmission probability of BS $\left(\tau_{BS}\right)$ in terms of $P_{c-BS}$ is given as follows: (2)$$\tau_{BS}=\frac{2}{1+W+P_{c-BS}W\sum_{i=0}^{M-1}{\left(2P_{c-BS}\right)}^{i}}.$$

Note that the expressions (Equation 1) and (Equation 2) serve as the starting point to compute QoS parameters for the Markov chain of RF-powered nodes. Equation (Equation 1), in practice, provides an upper bound on the collision probability of the BS involving RF-powered IoT nodes. Similarly, (Equation 2) provides a lower bound on the transmission probability of the BS. This is due to the fact that in a network with EH nodes, the total number of nodes participating in the competition is less than the *N* as some of them might not have sufficient energy.

The transmission probability of a node $\left(\tau_{N}\right)$, the collision probability of a node $\left(P_{c-N}\right)$, and the energy harvesting probability of a node $\left(P_{e}\right)$ are expressed as simultaneous Equations (Equation 3)–(Equation 5), along with (Equation 1) and (Equation 2). The QoS parameter expressions for an IoT node are obtained by simplifying the expressions provided [20] after the decoupling approximation. The transmission probability of a node $\left(\tau_{N}\right)$ is expressed as follows: (3)$$\tau_{N}=\frac{t_{2}-\sqrt{{t_{2}}^{2}-8t_{1}P_{e}}}{2t_{1}},$$
where
$$\begin{array}{cc} t_{1} & =(\left(W-1\right)P_{e}+P_{c-N}W\sum_{i=0}^{M-1}{\left(2P_{c-N}\right)}^{i})C_{th}\left(1-P_{c-N}\right),\\ t_{2} & =\left(W+1\right)P_{e}+P_{c-N}P_{e}W\sum_{i=0}^{M-1}{\left(2P_{c-N}\right)}^{i}+2C_{th}\left(1-P_{c-N}\right). \end{array}$$

The collision of a node occurs if some other node or the BS transmits when the node transmits. Hence, the collision probability of a node $\left(P_{c-N}\right)$ is expressed as follows: (4)$$P_{c-N}=1-\left(1-\tau_{BS}\right){\left(1-\tau_{N}\right)}^{\left(N-1\right)}.$$

The energy harvesting probability depends on the transmission and collision probability of the BS and is expressed as follows: (5)$$P_{e}=\frac{N-1}{N}\tau_{BS}{\left(1-\tau_{N}\right)}^{\left(N-1\right)}.$$

These expressions are used further to derive the remaining QoS parameters in Section 4.

## 4. Performance Analysis

In this section, we derive the important performance indices for a node in the network. The structure of an NR-U frame is discussed in Section 4.1. The closed-form expressions for the normalized saturated throughput of nodes and the BS are derived in Section 4.2. The mean delay at a node is calculated in Section 4.3. The outage probability is derived in Section 4.4.

### 4.1. The NR-U Frame Structure

The frame structure of the NR-U node with sufficient energy to transmit is depicted in Figure 6.

Short interframe spacing (SIFS) is similar to that in the IEEE 802.11b standard [5], where the node waits at the beginning of the frame to avoid collisions due to delayed acknowledgments (ACKs). In CCA duration, the node senses the channel for occupancy. This duration is typically 1 to 7 slots according to [4]. Next, if the channel is found to be free, the node draws a random value from the congestion window $\left(CW_{min},CW_{max}\right)$ as the backoff duration, denoted as BD. At the end of the backoff, the node transmits. If two or more nodes transmit, then it results in a collision; otherwise, it is a success. The transmission duration in the case of success $\left(T_{s}\right)$ is assumed to be the TXOP duration. The collision duration $\left(T_{c}\right)$ is assumed to be equal to one slot. Let δ be one slot duration. The average backoff duration $\left(\bar{BD}\right)$ in the network required for the computation of the throughput can be computed as follows.

In NR-U, the collision window is doubled after a collision, making the backoff process binary exponential, similar to carrier sense multiple access with collision avoidance (CSMA/CA). This can be modeled using exponential random variables, as described in [5]. Let $b_{1},b_{2},\ldots ,b_{N}$ be exponential random variables with mean β. The idle period in the network ends when one of the backoffs from *N* nodes finishes. This period is a minimum of *N* i.i.d. exponential random variables with mean β. This is, again, an exponential random variable with a mean of $\frac{\beta }{N}$. As the backoff, a node is uniformly distributed over $\left(CW_{min},CW_{max}\right)$, and the mean value of the backoff duration of a node (β) can be expressed as follows: (6)$$\beta =\frac{CW_{min}+CW_{max}}{4}.$$

Hence,
(7)$$\bar{BD}=\frac{CW_{min}+CW_{max}}{4N}\times \delta .$$

### 4.2. The Normalized Saturation Throughput

Next, we provide the analysis of the normalized saturation throughput. The normalized saturation throughput of nodes is denoted as $\theta_{N}$ and that of BS as $\theta_{BS}$. The throughput is saturated as each node always has a packet to transmit. Let $P_{Ts}$ be the probability of success and $P_{Tc}$ be the probability of collision. They are expressed as follows: (8)$$P_{Ts}=\tau_{BS}{\left(1-\tau_{N}\right)}^{N}+N\tau_{N}\left(1-\tau_{BS}\right){\left(1-\tau_{N}\right)}^{\left(N-1\right)},$$
(9)$$P_{Tc}=1-P_{Ts}.$$

Then, the normalized saturation throughput of nodes, which is the ratio of the expected time spent in successful transmission to the expected total time duration, is given as follows: (10)$$\theta_{N}=\frac{N\tau_{N}\left(1-\tau_{BS}\right){\left(1-\tau_{N}\right)}^{N-1}T_{s}}{P_{Ts}Ts+\left(1-P_{Ts}\right)T_{c}+T_{CA}},$$
where
$$\begin{array}{cc} T_{s} & =\mathrm{Time}\;\mathrm{spent}\;\mathrm{in}\;\mathrm{successful}\;\mathrm{transmission};\\ T_{c} & =\mathrm{Time}\;\mathrm{spent}\;\mathrm{in}\;\mathrm{collision};\\ T_{CA} & =SIFS+CCA+\bar{BD}. \end{array}$$

Next, $\theta_{BS}$ can be expressed as follows: (11)$$\theta_{BS}=\frac{\tau_{BS}{\left(1-\tau_{N}\right)}^{N}T_{s}}{P_{Ts}T_{s}+\left(1-P_{Ts}\right)T_{c}+T_{CA}}.$$

$\theta_{N}$ and $\theta_{BS}$ are simulated and analyzed for different parameters in Section 5.

### 4.3. Mean Delay of a Packet at a Node

In order to find the mean delay of a packet at an IoT node, the network with RF-powered NR-U nodes—except BS—is visualized as a queue sending data to BS. The queue’s service rate in terms of the number of packets transmitted per second is denoted as s(t) and the number of packets waiting in the queue at time *t* is denoted as N(t). Let $D_{i}$ be the delay of ith packet. We consider that a node transmits one packet whenever it has an opportunity during a successful transmission time Ts. As the Markov chain of Section 3.3 is a positive recurrent DTMC, it is stationary and ergodic. Hence, s(t) and $D_{i}$ are mean ergodic processes. Therefore,
$$\begin{array}{cc} \; & \frac{\int_{0}^{T}s(t).d(t)}{T}=\theta_{p}\;\mathrm{as}\;T\to \infty ,\\ \; & \frac{1}{n}\sum_{i=1}^{n}D_{i}=E\left[D\right]\;\mathrm{as}\;n\to \infty . \end{array}$$
where $\theta_{p}$ is throughput in terms of the mean number of packets transmitted per second. Then using Little’s theorem [31], we have the following: (12)$$E\left[N\left(t\right)\right]=\theta_{p}E[D].$$

Under the saturated network condition, E[N(t)] is *N*. Hence,
(13)$$E\left[D\right]=\frac{N}{\theta_{p}}.$$

$\theta_{p}$ is $\theta_{N}\setminus T_{s}$. Therefore,
(14)$$E\left[D\right]=\frac{NTs}{\theta_{N}}.$$

The derived results in this section are validated using simulations in Section 5.

### 4.4. Outage Probability of the Node

The charging level of the supercapacitor, i.e., the energy level of the node c(t) can be represented by the Markov chain shown in Figure 7.

The charging level is incremented by one from 0 to $C_{max}$ with probability $P_{e}$. The energy of the node reduces to zero whenever a successful transmission occurs for states with energy greater than or equal to $C_{th}$. The probability of successful transmission for a node denoted as $q_{N}$ is $\tau_{N}\left(1-P_{c-N}\right)$. The stationary distribution of the Markov chain in Figure 7 can be computed by solving $\pi =\pi P_{E}$ where $P_{E}$ is the transition matrix of the Markov chain.

The outage probability of a node is the probability that the node does not have the energy to transmit the packet even though the packet is available at the node. Due to the assumption of saturated conditions, the outage probability of the node is the probability that the energy level of the node is less than $C_{th}$. Let $P_{o}$ be the outage probability of the node, which is represented as follows: (15)$$P_{o}=\sum_{i=0}^{C_{th}-1}Pr\left(c\left(t\right)=i\right).$$

The outage probability is computed and analyzed in Section 5.

## 5. Simulations

In this section, the simulation results for the QoS parameters of the RF-powered NR-U network described in Section 3 and Section 4 are presented. To check the validity of the mathematical model, a Monte Carlo simulator involving essential functions of an RF-powered LBT network is built in the programming language C. The transmit power of the base station (*Q*) is considered 1 W. The radius of the circular area is considered to be 9 m. The distances and power of the base station are considered according to the energy harvesting requirement of the nodes similar to [20] and also the 3GPP regulatory requirements on transmit power [4]. These distances are suitable for ISM bands (2.4 GHz and 5–7 GHz). For the mmWave band, the distances will be shorter, nonetheless, the model will be applicable. The values of simulation parameters are listed in Table 2. The number of nodes varied from 1 to 100 according to the requirements of different IoT applications, e.g., an agricultural IoT network has a size of around 100 nodes. The congestion window size combinations and TXOP parameter values are chosen according to the 3GPP guidelines provided [4].

In Figure 8, the collision probability and the transmission probability of the node are plotted with respect to the number of nodes. As the number of nodes increases, the collision probability of the node increases, and the transmission probability decreases exponentially as more nodes compete simultaneously.

In Figure 9, the energy harvesting and outage probabilities of the IoT node are plotted. The energy harvesting probability of the node depends on the transmission probability of the BS, hence, it decreases with the increase in the number of nodes. As the number of nodes increases further, there is no significant change in the energy harvesting probability and the outage probability, indicating the energy sustainability of the network. The gap between theoretical and simulation values for fewer than 40 nodes is due to the fact that BS expressions (Equation 1) and (Equation 2) provide bounds instead of exact values. As the number of nodes increases, this effect reduces.

The total throughput of nodes for different congestion window sizes is shown in Figure 10.

The analytical model matches well with the simulations. It can be seen that there is no change in the throughput for the congestion window size.

In Figure 11, the throughput for nodes, as well as the BS for different values of the TXOP parameter, are plotted. The simulation values of throughput for TXOP values 4 ms, 6 ms, and 8 ms are shown for the congestion window sizes (16,128). Changing the TXOP value has very little effect on the throughput.

The mean delay of a packet at a node is shown for different values of the number of nodes in the network in Figure 12.

As the number of nodes increases, the competition among nodes increases. Hence, nodes have a lesser chance to transmit, increasing the mean delay of a packet at a node linearly. The change in the mean delay at a node for the TXOP parameter is shown in Figure 13. As the TXOP parameter value increases, the mean delay increases.

The effect of changing $C_{max}$, which is the energy storage capacity of the node, is observed in Figure 14. As seen in the theoretical analysis of Section 3, the performance of the node is independent of the $C_{max}$.

**Figure 12 sensors-24-05369-f012:**
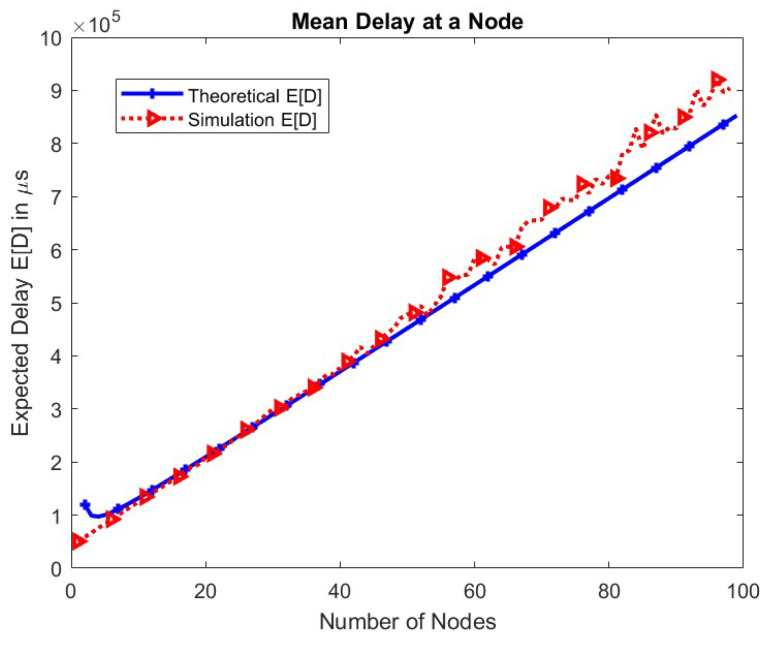
Mean delay of a node.

**Figure 13 sensors-24-05369-f013:**
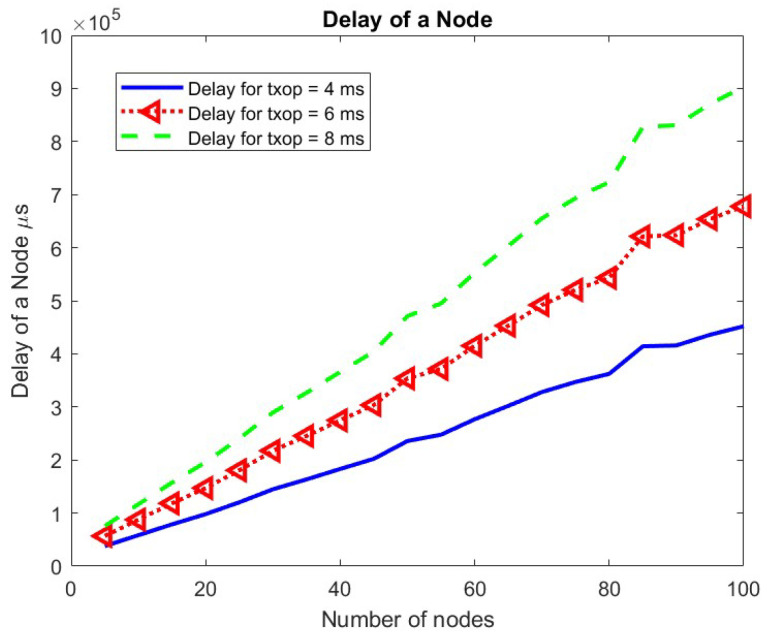
Mean delay at a node w.r.t. TXOP.

**Figure 14 sensors-24-05369-f014:**
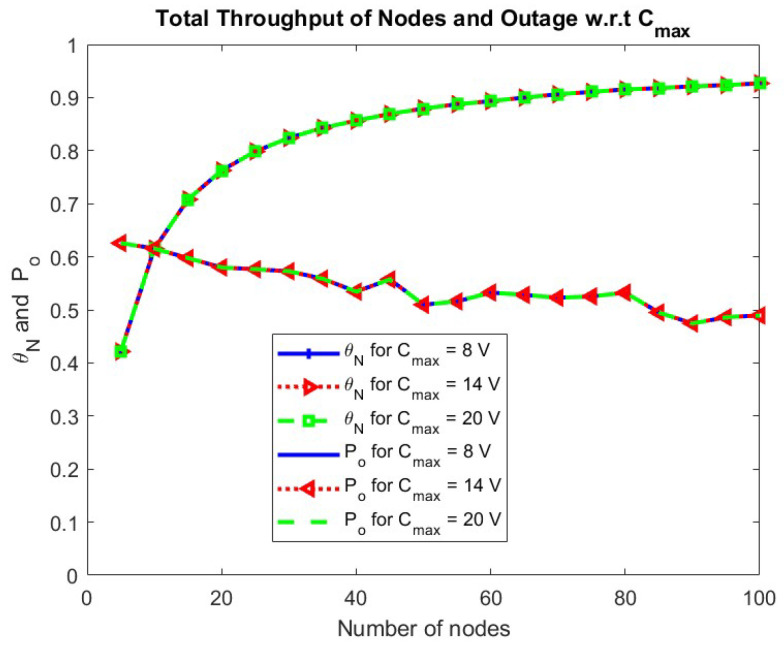
Total throughput of nodes and outage w.r.t. $C_{max}$.

The effect of changing the energy threshold $C_{th}$ on the throughput of nodes and the mean delay is observed in Figure 15 and Figure 16, respectively. The throughput of nodes decreases, while the mean delay of the node remains unchanged with an increase in the minimum energy required for operation.

**Figure 15 sensors-24-05369-f015:**
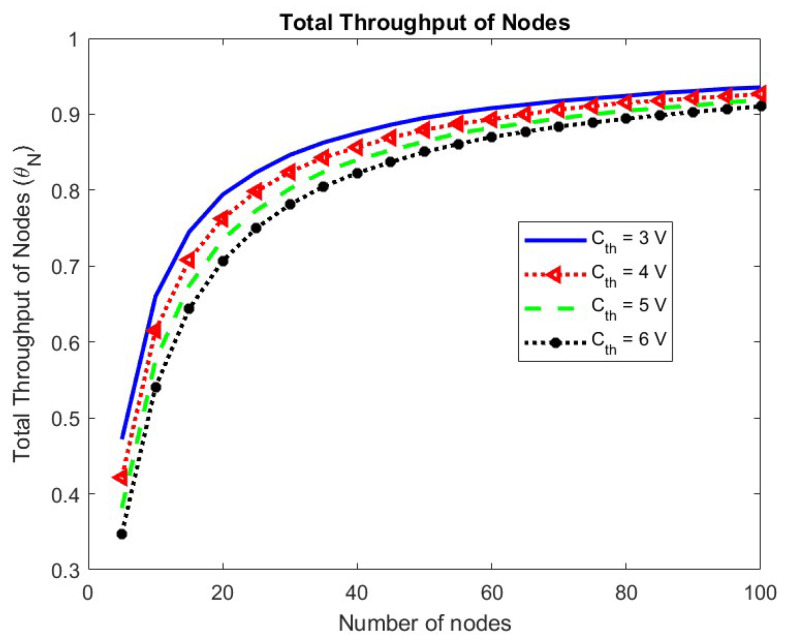
Total throughput of nodes w.r.t. $C_{th}$.

**Figure 16 sensors-24-05369-f016:**
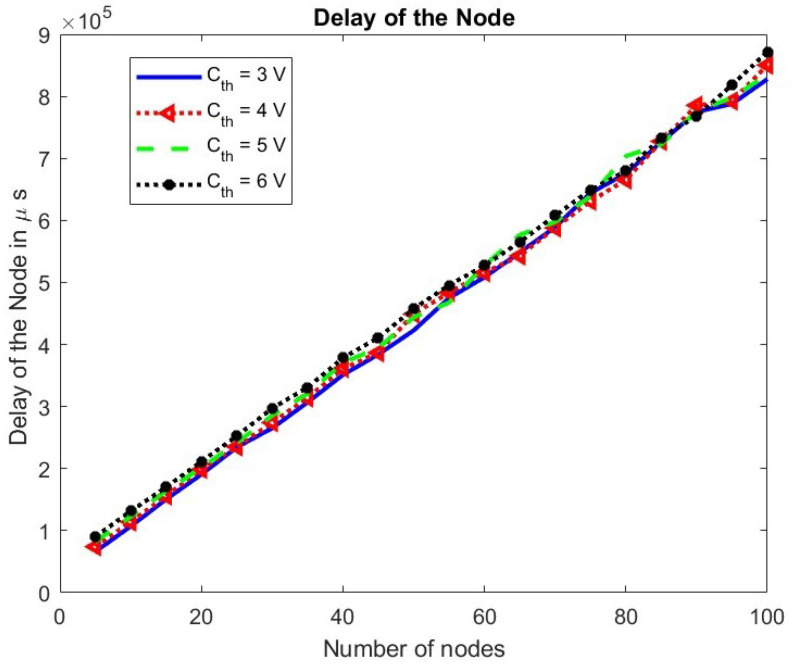
Delay at a node w.r.t. $C_{th}$.

The outage probability of the node increases with an increase in the energy threshold $C_{th}$ as seen in Figure 17.

The effect of change in $C_{th}$ on the collision and transmission probabilities of the node and the BS is shown in Figure 18 and Figure 19, respectively. It is seen that the transmission probability of the node remains the same for change in $C_{th}$ while the transmission probability for the BS increases. This is because fewer nodes become eligible for participation due to increased $C_{th}$ giving more chance to BS. An increase in the BS transmissions results in more harvested energy, compensating for the increased value of the minimum energy required for operation. Consequently, the transmission probability of the node remains unchanged. The collision probability for both the node and the BS decreases.

**Figure 17 sensors-24-05369-f017:**
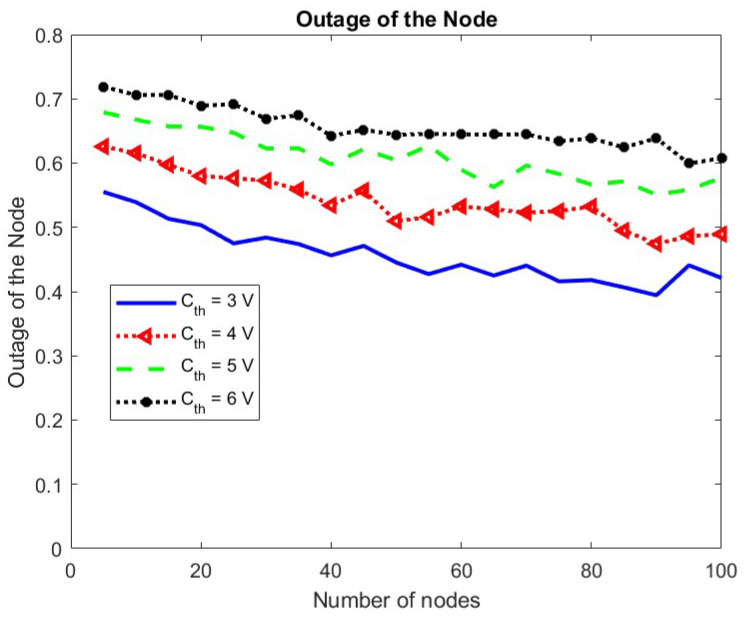
Outage probability of node w.r.t. $C_{th}$.

**Figure 18 sensors-24-05369-f018:**
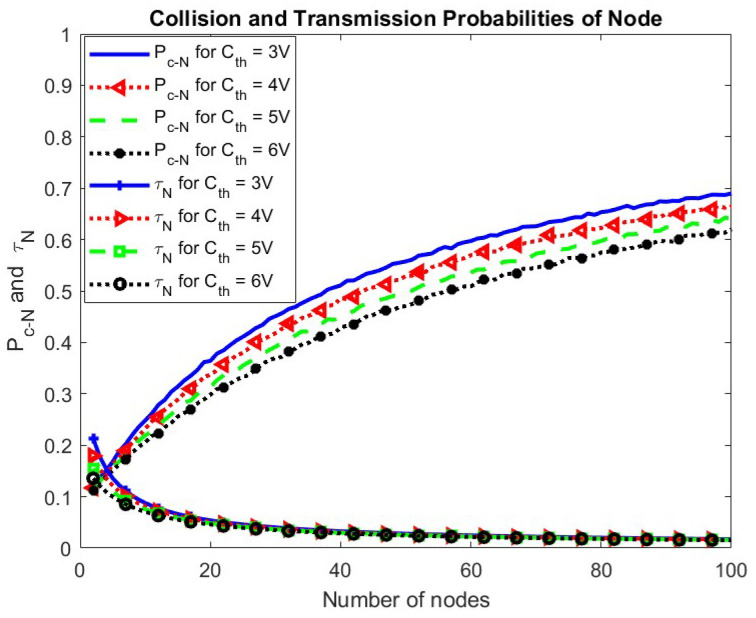
Collision and transmission probabilities of node w.r.t. $C_{th}$.

**Figure 19 sensors-24-05369-f019:**
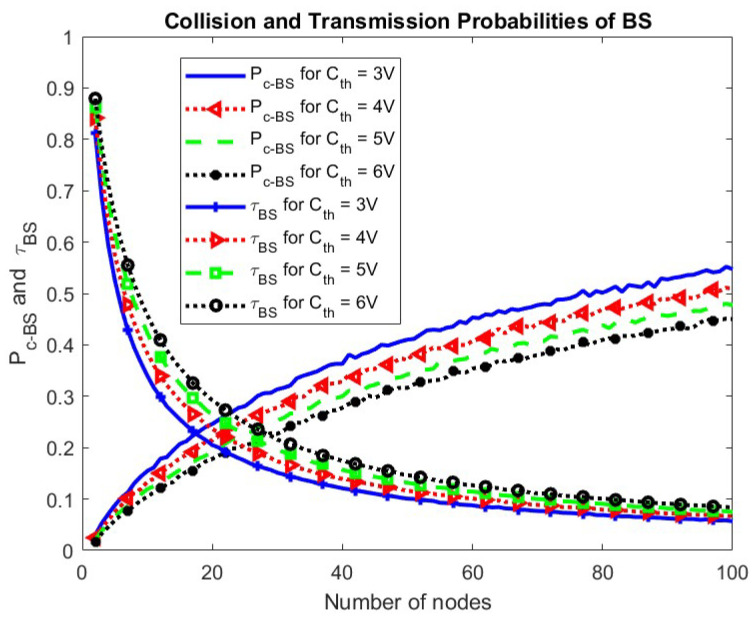
Collision and transmission probabilities of BS w.r.t. $C_{th}$.

## 6. Conclusions

We modeled an NR-U network with RF energy harvesting nodes in this work using a three-dimensional Markov chain. The closed-form expressions for the normalized saturated throughput of the IoT node and the BS, as well as the mean delay of a packet at the node, are derived. In addition, the node’s transmit outage probability is computed. The collision and transmission probabilities of the node and the BS, along with the energy harvesting probability of the node, are analyzed based on the network’s density. The theoretical values for QoS parameters align well with simulations. The collision probability increases and the transmission probability decreases for both the node and the BS as the number of nodes increases. The node’s energy harvesting probability decreases with an increase in the number of nodes. The throughput of the BS decreases, while the total throughput of nodes increases with an increase in the number of nodes. The mean delay is observed to increase linearly with the density of the network. The outage probability of the node remains the same even though the number of nodes increased.

The effects of different congestion window sizes on the throughput and delay are observed. The change in TXOP value does not have a significant impact on the throughput but the mean delay changes linearly. As seen in the theoretical expressions and corresponding simulations, the throughput, delay, and outage are independent of the maximum energy level of a node. The throughput and outage probability depend on the energy threshold rather than the maximum energy storage level of the node. The delay is independent of the energy threshold. Thus, a theoretical framework for an RF-powered IoT network following the LBT protocol is provided, and QoS parameters are analyzed. The closed-form expressions presented in this work can be used to optimize the QoS parameters for system variables of interest. This framework is useful for analyzing the performance of LBT-based RF-powered IoT networks used in diverse areas, including agriculture, industrial, and healthcare.

## 7. Future Work

The model can be generalized to consider different EH requirements of each node. Also, different energy storage capacities and thresholds can be incorporated into the model. In this work, homogeneous coexistence is considered. The model can be extended to take into account the existence of Wi-Fi (either RF-powered or traditional), although the complexity will increase.

## Figures and Tables

**Figure 1 sensors-24-05369-f001:**
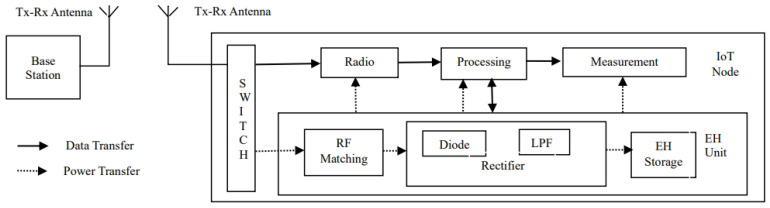
A typical node with an RF energy harvester.

**Figure 2 sensors-24-05369-f002:**
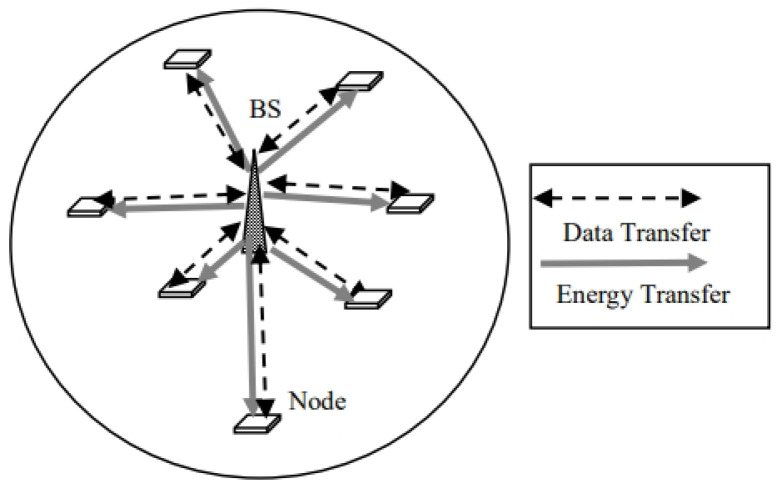
System model.

**Figure 3 sensors-24-05369-f003:**
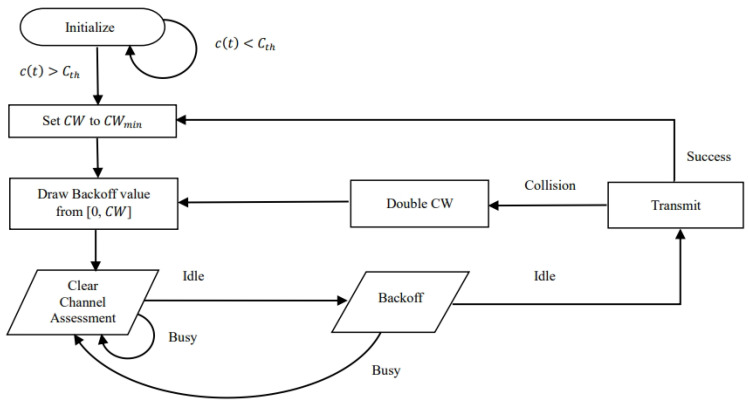
LBT procedure of RF-powered NR-U node.

**Figure 4 sensors-24-05369-f004:**
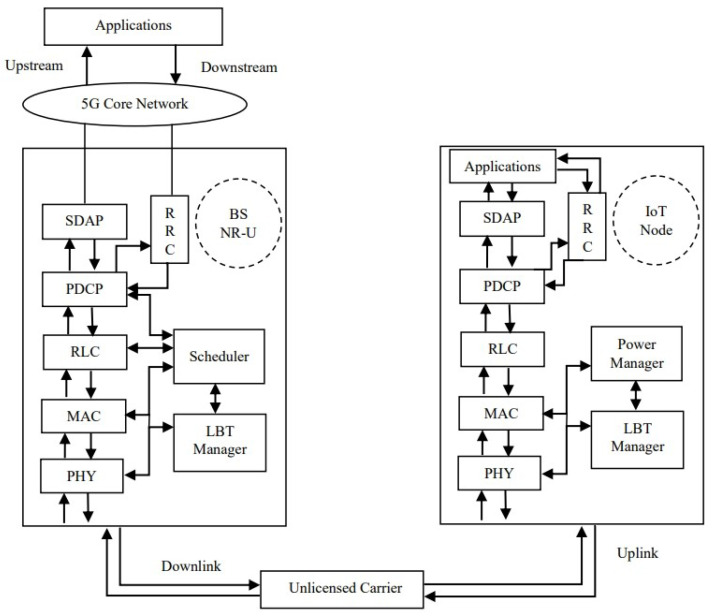
Protocol stack of the RF-powered NR-U system.

**Figure 5 sensors-24-05369-f005:**
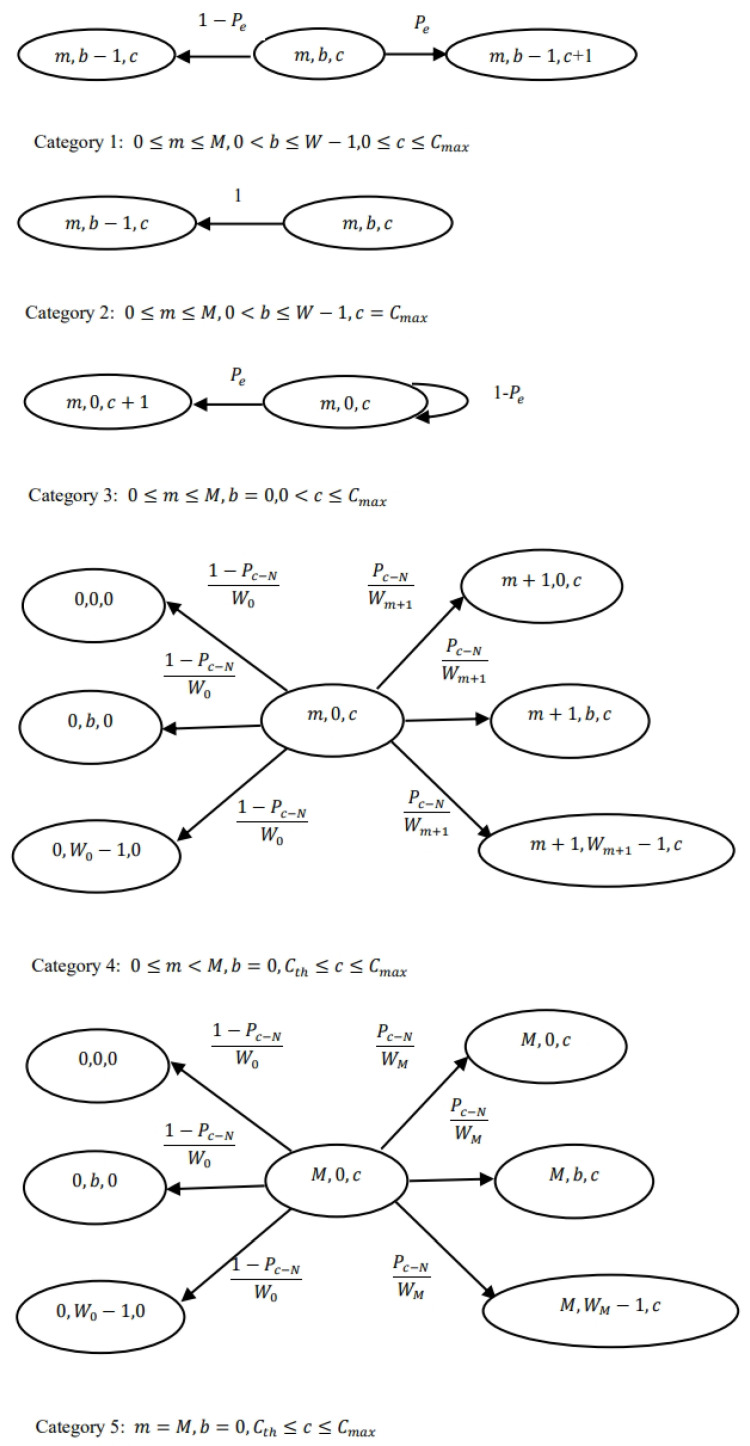
State space diagram.

**Figure 6 sensors-24-05369-f006:**
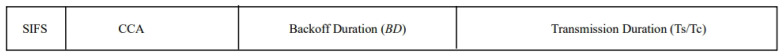
Frame Structure of the NR-U Node.

**Figure 7 sensors-24-05369-f007:**
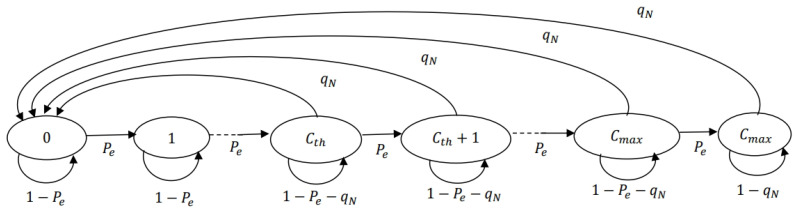
Markov chain of the charging level of the node.

**Figure 8 sensors-24-05369-f008:**
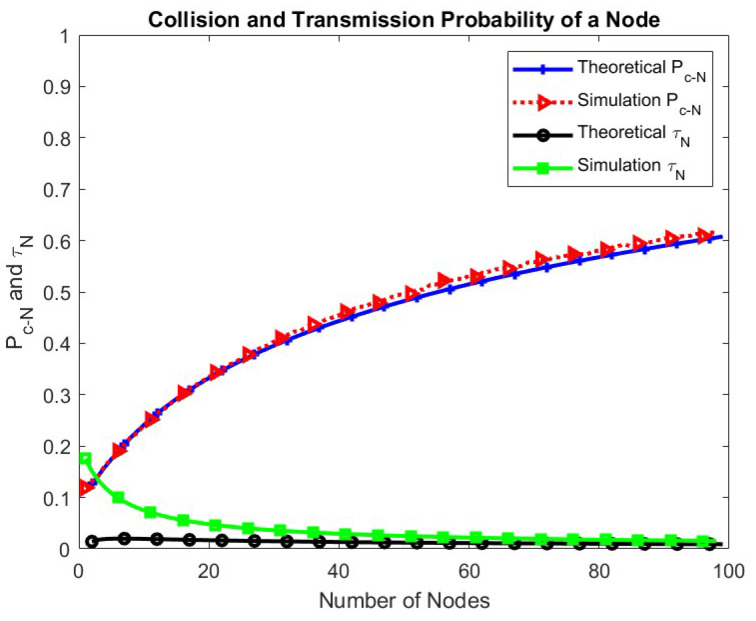
Collision and transmission probabilities of a node.

**Figure 9 sensors-24-05369-f009:**
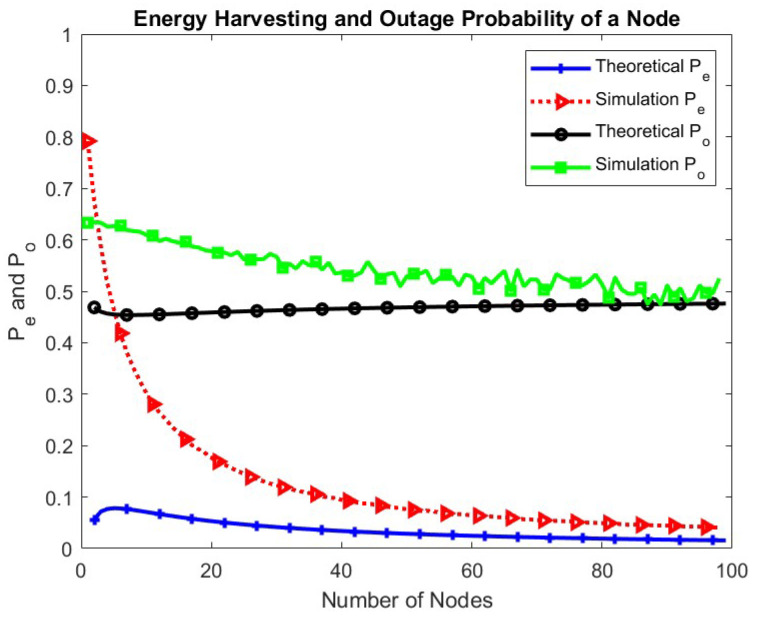
Energy harvesting and outage probabilities of a node.

**Figure 10 sensors-24-05369-f010:**
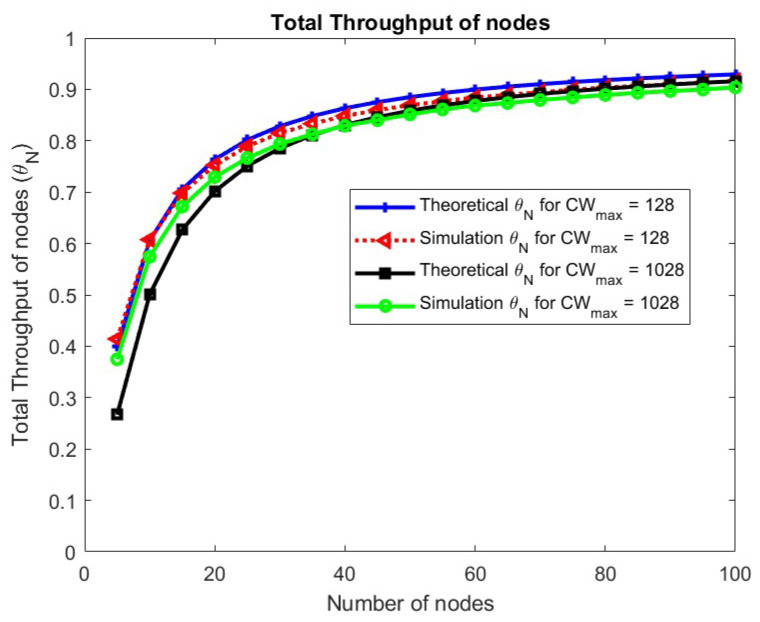
Total throughput of nodes w.r.t. $CW_{max}$.

**Figure 11 sensors-24-05369-f011:**
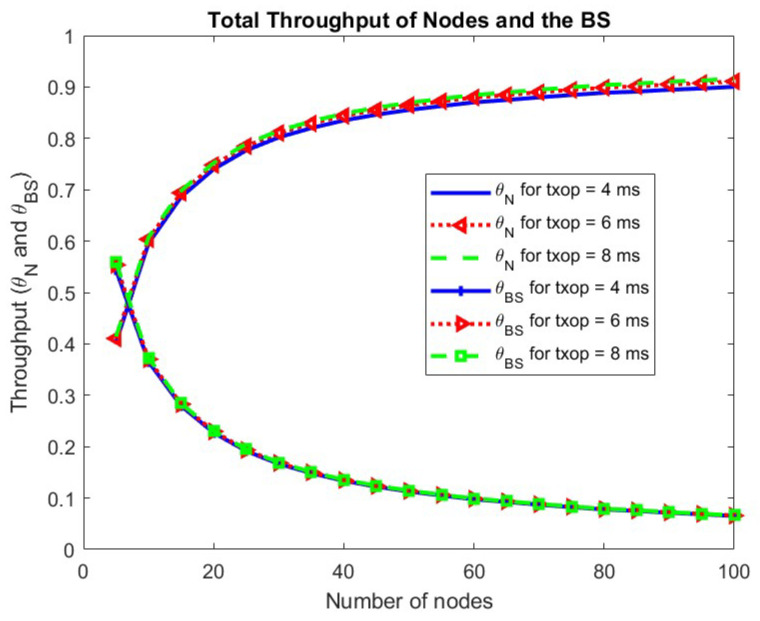
Total throughput of nodes and BS w.r.t. TXOP.

**Table 1 sensors-24-05369-t001:** Notations.

Notation	Description
*Q*	The transmit power of the BS
*R*	The radius of the circular area
$C_{th}$	Threshold for energy level above which a node can transmit
$C_{max}$	Maximum value of the energy level of the supercapacitor
CW	Congestion window size
$CW_{min}$	Minimum value of the congestion window
$CW_{max}$	Maximum value of the congestion window
*N*	Number of stations
$W_{0}$	Initial congestion window size
$W_{m}$	Congestion window size after $m^{th}$ retransmission attempt
*M*	Maximum number of times the congestion window can be doubled
m(t)	Number of times the congestion window doubled until time *t*
b(t)	Backoff counter value at time *t*
c(t)	Charging level of supercapacitor at time *t*
$P_{e}$	Probability that a node successfully harvests the energy from BS
$\tau_{N}$	Probability that a node attempts a transmission
$P_{c-N}$	Probability that a node collides with another node
$P_{c-BS}$	Probability that BS encounters a collision
$\tau_{BS}$	Transmission probability of BS
$P_{e}$	Energy harvesting probability of the node
*P*	Probability transition matrix of the Markov chain
$\theta_{N}$,$\theta_{BS}$	Normalized saturation throughput of nodes and BS, respectively
$P_{Ts}$	Probability of the success of a node
$P_{Tc}$	Probability of the collision of a node
$\bar{BD}$	Average backoff duration
δ	One slot duration
E[D]	Expected delay of a node
$N_{p}$	Mean number of packets waiting for transmission at the front of the queue in the network
$\theta_{p}$	Throughput in terms of the mean of packets per second
$T_{s}$	Transmission time in case of success
$P_{o}$	Probability of outage of a node

**Table 2 sensors-24-05369-t002:** Simulation parameters.

Parameters	Figure 8 and Figure 9	Figure 10	Figure 11	Figure 12	Figure 13	Figure 14	Figure 15, Figure 16, Figure 17, Figure 18 and Figure 19
$CW_{max}$	128	128,1028	128	128	128	128	128
TXOP (in ms)	8	8	4, 6, 8	8	4, 6, 8	8	8
$C_{max}$ (in Volts)	8	8	8	8	8	8,14,20	8
$C_{th}$ (in Volts)	4	4	4	4	4	4	3,4,5,6
$CW_{min}$	16						
No. of Nodes	1–100						
SIFS (in μs)	16						
CCA (in μs)	63						
δ (in μs)	9						

## Data Availability

The original contributions presented in the study are included in the article.

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
