# Peer review of "Modeling and Performance Analysis of LBT-Based RF-Powered NR-U Network for IoT"

_sensors, 2024, doi:10.3390/s24165369_

Round 1

Reviewer 1 Report

Comments and Suggestions for Authors

1. The Abstract should re-written again. Please, explain; why this research?

2. Regardingthe performance analysis, in addition to the impact of LBT on the REF Energu Harvesting, what about the coexistence with other technologies, i.e., Assess the impact of interference from other unlicensed spectrum users on the LBT process and overall network performance (Interference analysis). Please explain.

Comments on the Quality of English Language

There are some grammar issues that should be considered by the author, e.g., in the Abstract; The collision probability; the transmission probability and the energy harvesting probability of the nodes is analyzed with respect to the density of network. "is" must be "are".

Author Response

1.The Abstract should re-written again. Please, explain; why this research?

Thank you for pointing this out. We agree with this comment. Therefore, the abstract is rewritten. The need for the research work performed in this paper is mentioned. The revised abstract is on Page no.1 and starting with line no.1.

  1. Regarding the performance analysis, in addition to the impact of LBT on the REF Energu Harvesting, what about the coexistence with other technologies, i.e., Assess the impact of interference from other unlicensed spectrum users on the LBT process and overall network performance (Interference analysis). Please explain.

Thank you for pointing this out. We agree with this comment. In this model, the coexistence with WiFi is not considered. This is called as homogeneous coexistence. It is assumed that some kind of multiplexing either spatial or time is used to avoid the interference from technologies in unlicensed band other than WiFi such as Bluetooth. The model can be extended to take into account the existence of WiFi (either RF powered or traditional) though the complexity will be high. Also, the Signal to Interference ratio (SINR) can be incorporated in throughput calculations as part of Future Work. The explanation is added in Section 3 on Page no. 4- line no. 163 and in Conclusions on Page no. 17-line no. 408

Comments on the Quality of English Language

There are some grammar issues that should be considered by the author, e.g., in the Abstract; The collision probability; the transmission probability and the energy harvesting probability of the nodes is analyzed with respect to the density of network. "is" must be "are".

Thank you for pointing this out. We agree with this comment. Hence, the article is modified to correct English language discrepancies.

Reviewer 2 Report

Comments and Suggestions for Authors

In this paper, the author  investigates a sustainable cellular network for the Internet of Things (IoT) called an NR-U network, where nodes are powered by a base station using radio frequency (RF) energy harvesting. A list of points that deserve to be better clarified in the paper and some suggestions follows.

1. The manuscript is technically weak.

2. The abstract part of this article is more complex, making it difficult to understand the main idea. So, revise the abstract.

3. More details literature review is needed to clarify about work.

4. More details are needed to clarify the structure of the model and implementation in sections 3.

5. In equation 1 to 14 did not use any kind of reference. How to justify these equations are valid?

6. The performances of analysis of models are developed. How to prove good stability and generalization.

7. How the proposed NN framework towards approaching the global optimal result?

8. The authors should refer in more detail and clarity to the advantages of their methodology and summarize possible limitations.

Author Response

  1. The manuscript is technically weak.

Thank you for pointing this out. We agree that in some parts it appeared to be weak.Hence, the manuscript is modified according to reviewer’s comments to increase the credibility. The mistake regarding nomenclature gNB used is corrected (Page no.4, line no. 157). More diagrams and explanations are added (Fig. 1, Fig.2 and Fig.3). Also, the references required for explanation are added. The analysis of simulation graphs is enhanced. The diagram of outage probability Markov chain is corrected to take into account missing terms (Page no. 11, Fig 6).

  1. The abstract part of this article is more complex, making it difficult to understand the main idea. So, revise the abstract.

Thank you for pointing this out. We agree with this comment. Therefore, the Abstract is rewritten to convey the main idea and work done in proper sentences and flow (Page no.1 and line no.1).

  1. More details literature review is needed to clarify about work.

Thank you for pointing this out. We agree with this comment. Hence, the literature review is updated. The references regarding the outage probability are added (Page no. 3, line no. 130). The author’s contributions are elaborated in the Section 2 (Page no. 4, line no. 143)

  1. More details are needed to clarify the structure of the model and implementation in sections 3

Thank you for pointing this out. We agree with this comment. Hence, the explanation with block diagram and related references are added for RF Energy harvester unit (Page no. 4, line no. 167) and protocol stack(Page no. 6, line no. 202). The system model figure is updated in order to bring clarity (Page no.5, Fig. 2). The gNB nomenclature was used for an IoT node instead of base station in earlier manuscript version. This mistake is corrected. (Page no.4, line no. 157).

  1. In equation 1 to 14 did not use any kind of reference. How to justify these equations are valid?

Thank you for pointing this out. We agree with this comment. Equation 1 to 5 are in the form suitable for the system model considered in this paper. They are simplified from the results derived after decoupling approximation in the reference cited. Equation 6 to 14 are derived in this paper using equation 1 to 5. Explanation is added in section 3.4 (Page no. 7, line no. 256, line no. 268).

  1. The performances of analysis of models are developed. How to prove good stability and generalization.

Thank you for pointing this out. We agree with this comment. The performance analysis is performed by modelling by a Markov chain which is positive recurrent hence has steady state distribution indicating the stability of the mathematical model. The explanation is available at (Page no.7, line no.227). The system model i.e., the RF powered NR-U network with IoT nodes can be generalized for each node having different energy thresholds and different energy harvested in one time slot and different capacity of the supercapacitor energy storage. The explanation is added at (Page no.17, line no. 408)

  1. How the proposed NN framework towards approaching the global optimal result?

Thank you for the comment. The global optimal term is part of the optimization technique. I answer this in the Markov chain context with respect to stability.

  1. Mathematical Model: The Markov chain of the model under consideration is stable. (Page no.7, line no.227). It provides us with expressions for QoS parameters. Applying optimization techniques for QoS parameters according to system variables can be performed as part of the Future Work. The explanation is added in Conclusions (Page no. 17, Line no. 408)
  2. System Model: The scheme considered here is the simplest without any significant changes to standardized functioning of NR-U or Base Station. Many papers target towards improving the NR-U algorithm in non-EH setup. These schemes can be evaluated for EH setup as part of Future Work. 

8. The authors should refer in more detail and clarity to the advantages of their methodology and summarize possible limitations.

Thank you for pointing this out. We agree with this comment. The advantages and limitations are added to Conclusions (Page no. 17, Line no. 408). Also, at the places where the explanation is relevant e.g. in Section 3 while explaining the system model (Page no.4, line no. 180).

Round 2

Reviewer 2 Report

Comments and Suggestions for Authors

Still, the main problem of the paper is that it is difficult to understand. The contributions still need to be clarified and lack novelty.

It would be best to compare with at least some of the papers mentioned in Section 1. Otherwise, no matter how novelty is your proposed method.

Why choose the three-dimensional Markov Chain model? Why not another model?

What about the Energy-Efficient?

Too few points to draw that Result Section, Comparison with other methods (even if in different fields)? Mathematical support? Can benefit from more experimental data.

Author Response

Thank you for going through the manuscript and providing reviews. Here is my response to your comments.

  1. Still, the main problem of the paper is that it is difficult to understand.

Response:  Thank you for pointing this out. We agree that the main algorithm needed some more explanation. Hence, the flowchart showing the LBT procedure for a RF powered NR-U node is added in Section 3.2 (Page 5, line no. 194, Fig. 3).

  1. The contributions still need to be clarified and lack novelty.

It would be best to compare with at least some of the papers mentioned in Section 1. Otherwise, no matter how novelty is your proposed method.

Response: Thank you for the comment. The author’s contribution is highlighted.  The authors contribution with respect to non-EH NR-U network analysis of reference [19] is added in Section 2 at the end of the Literature Survey (Page no. 4, line no. 148). The NR-U with energy harvesting is envisioned first time in this work. The authors propose to use NR-U with energy harvesting and further analyze it for performance. Hence comparison with RF powered IoT network using communication protocol other than LBT is not possible. Also, NR-U with traditionally powered IoT node cannot be directly compared with for values of QoS parameters.

  1. Why choose the three-dimensional Markov Chain model? Why not another model?

Response: Thank you for pointing this out. We agree with the comment. Hence, the explanation for benefit of Markov modelling (Page no.6, line no. 228) and reason for using three-dimensional Markov chain (Page no. 7, line no. 238) is added Section 3.3.

  1. What about the Energy-Efficient?

Response: Thank you for the comment. The focus of the paper is on modelling and getting closed form expressions for LBT (standardized by 3GPP for non-EH) in an energy harvesting scenario without significant changes. These expressions are further analyzed for some important parameters.  The modifications in current procedure for energy efficiency are open for further research.

  1. Too few points to draw that Result Section, Comparison with other methods (even if in different fields)? Mathematical support? Can benefit from more experimental data.

 Response: Thank you for the comment. We agree that few more results will help. Hence, more experimental data is added and analyzed.  The simulation results (Fig.20 to Fig. 23) are added in Section 5 for variable energy threshold (Page no. 16, line no. 407). Conclusions are rewritten (Page no. 19, line no. 415). The generalizations which need extensive change in current mathematical model are moved to Future Work section (Page no. 19, line no. 440). The direct comparison with other methods for values of QoS parameters is not possible as the RF powered NR-U is put forth first time in this paper and not standardized yet by any regulating body.
